# MCM4 as Potential Metastatic Biomarker in Lung Adenocarcinoma

**DOI:** 10.3390/diagnostics15121555

**Published:** 2025-06-18

**Authors:** Hung-Chih Lai, Ju-Fang Liu, Tsung-Ming Chang, Thai-Yen Ling

**Affiliations:** 1Department of Hematology and Oncology, Shin Kong Wu Ho-Su Memorial Hospital, Taipei 111045, Taiwan; m012584@ms.skh.org.tw; 2Graduate Institute of Pharmacology, National Taiwan University College of Medicine, Taipei 100225, Taiwan; 3Translational Medicine Center, Shin-Kong Wu Ho-Su Memorial Hospital, Taipei 111045, Taiwan; jufangliu@tmu.edu.tw; 4Department of Medical Research, China Medical University Hospital, China Medical University, Taichung 404271, Taiwan; 5School of Oral Hygiene, College of Oral Medicine, Taipei Medical University, Taipei 110301, Taiwan; 6School of Dental Technology, College of Oral Medicine, Taipei Medical University, Taipei 110301, Taiwan; a03441@tmu.edu.tw; 7Institute of Pharmacology, College of Medicine, National Taiwan University, Taipei 100225, Taiwan

**Keywords:** lung adenocarcinoma, differentially expressed genes, metastasis, MCM4

## Abstract

**Background:** Lung adenocarcinoma (LUAD) is the most common subtype of non-small-cell lung cancer and is frequently diagnosed at advanced stages with metastasis, contributing to its poor prognosis. Identifying key metastasis-related biomarkers is critical for improving early diagnosis and therapeutic targeting. **Methods:** We analyzed four GEO microarray datasets (GSE32863, GSE27262, GSE40275, and GSE33356) and TCGA data to identify differentially expressed genes (DEGs) in LUAD. Functional enrichment of DEGs was analyzed using Gene Ontology, Kyoto Encyclopedia of Genes and Genomes analysis, and a Cancer Hallmark Enrichment Plot. Hub gene analysis was conducted using Cytoscape. Hub genes were evaluated for their expression, prognostic significance (via the Kaplan–Meier plotter), and clinical correlation using additional platforms (TCGA, Lung Cancer Explorer, TNMplot, and the Human Protein Atlas). **Results:** A total of 333 consistently dysregulated DEGs were identified, enriched in pathways related to metastasis, including angiogenesis, immune escape, and ECM interaction. Ten hub genes (*AURKA*, *TOP2A*, *CCNB2*, *CENPF*, *MCM4*, *TPX2*, *KIF20A*, *ASPM*, *MELK*, and *NEK2*) were identified through network analysis. Among these, *MCM4* showed strong upregulation in LUAD and was significantly associated with poor overall survival. Notably, *MCM4* expression also correlated with post-progression survival and markers of invasiveness. Immunohistochemistry and transcriptomic analyses confirmed *MCM4* overexpression at both mRNA and protein levels. Additionally, *MCM4* expression was positively correlated with various matrix metalloproteinases, supporting its role in promoting tumor invasiveness. **Conclusions:** *MCM4* is a novel potential biomarker for LUAD metastasis and prognosis. Its consistent upregulation, association with metastatic markers, and clinical significance suggest it may serve as a candidate target for diagnostic use or therapeutic intervention in advanced LUAD.

## 1. Introduction

Lung cancer is the leading cause of cancer-related mortality worldwide, with over two million new cases and 1.8 million deaths reported annually [1]. Non-small-cell lung cancer (NSCLC) accounts for approximately 85% of all lung cancer cases, with lung adenocarcinoma (LUAD) being the most prevalent histological subtype [2]. Despite advancements in surgery, radiotherapy, chemotherapy, targeted therapy, and immunotherapy, the five-year survival rate for LUAD remains dismally low, especially in patients diagnosed at advanced stages [3]. Among various clinical factors, tumor metastasis is the principal cause of treatment failure and poor prognosis, severely limiting the effectiveness of therapeutic interventions [4]. Therefore, identifying novel genes that contribute to metastasis is critical for improving early detection, prognostic evaluation, and the development of targeted therapies in LUAD.

Metastasis is a hallmark of cancer progression and is closely associated with increased malignancy and shortened survival [4,5]. It involves a cascade of biological events, including epithelial–mesenchymal transition (EMT), extracellular matrix remodeling, cell motility, and invasion into distant organs. The minichromosome maintenance (MCM) protein family, especially the MCM2–7 complex, plays a central role in DNA replication licensing and cell cycle regulation [6]. Dysregulation of MCM proteins has been implicated in genomic instability and uncontrolled proliferation—two key features of cancer development. Among these, MCM4 functions as a core subunit of the replicative helicase and is essential for unwinding DNA at replication origins [7]. Emerging evidence suggests that aberrant *MCM4* expression is associated with tumorigenesis and poor prognosis in several malignancies. For example, *MCM4* overexpression promotes cell proliferation and is associated with adverse outcomes in gliomas, endometrial cancer, hepatocellular carcinoma, and colorectal cancer [8,9,10,11]. In uterine corpus endometrial carcinoma, *MCM4* has been proposed as a novel prognostic biomarker based on its correlation with disease progression [9,12]. In glioma, high *MCM4* expression facilitates tumor cell growth and is linked to reduced survival [8]. Similarly, *MCM4* contributes to aggressive phenotypes in colorectal and liver cancers, including increased invasiveness and resistance to apoptosis [13,14]. However, the biological role and clinical significance of *MCM4* in lung adenocarcinoma remain largely unexplored, and whether it plays a role in LUAD metastasis has not been systematically evaluated.

In this study, we employed integrated bioinformatics analysis using publicly available LUAD transcriptome datasets to identify candidate genes associated with tumor metastasis and poor prognosis. Through differential expression analysis, survival correlation, and validation in clinical specimens, we identified *MCM4* as a significantly overexpressed gene in LUAD that is associated with lymph node involvement and reduced overall survival. Together, this study provides novel evidence that *MCM4* may serve as a potential biomarker for LUAD metastasis and a prognostic indicator for patient survival. These findings contribute to the growing understanding of replication-related gene dysregulation in cancer and highlight *MCM4* as a promising molecular target in the context of lung adenocarcinoma.

## 2. Materials and Methods

### 2.1. Data Acquisition

Four publicly available gene expression microarray datasets—GSE32863, GSE27262, GSE40275, and GSE33356—were obtained from the Gene Expression Omnibus (GEO) database (GEO; https://www.ncbi.nlm.nih.gov/geo/) (accessed on 16 April 2025) to identify differentially expressed genes (DEGs) between lung cancer and adjacent normal lung tissues: GSE32863 (18 February 2019) was generated using the Illumina HumanWG-6 v3.0 expression BeadChip (GPL6884) platform. This dataset includes 58 paired lung adenocarcinoma tumors and matched adjacent non-tumor lung tissues, derived from fresh-frozen specimens. The gene expression profiles were integrated with DNA methylation data in the original study to explore epigenetic regulation in LUAD. GSE27262 (25 March 2019) was produced using the Affymetrix Human Genome U133 Plus 2.0 Array (GPL570) platform. It contains 25 matched pairs of tumor and adjacent normal lung tissues obtained from stage I lung adenocarcinoma patients in Taiwan. GSE40275 (4 November 2020) utilized the Affymetrix Human Exon 1.0 ST Array (GPL15974) platform with Brainarray CDF annotation (HsEx10stv2_Hs_REFSEQ). This dataset includes 8 non-small-cell lung cancer (NSCLC) and 43 normal lung tissue samples, with RNA samples purchased from OriGene Technologies. GSE33356 (15 January 2020) was also analyzed on the Affymetrix Human Genome U133 Plus 2.0 Array (GPL570). It includes 60 matched tumor and adjacent normal lung tissue pairs from lung cancer patients in Japan [15,16,17]. These datasets were selected based on their adequate sample sizes, availability of matched tumor–normal pairs, and the use of standardized microarray platforms. Their integration allowed for robust cross-validation of DEGs and reliable biological interpretation, as described in Figure 1.

### 2.2. Filtration of Differentially Expressed Genes

Differentially expressed genes (DEGs) in lung cancer were identified using the GEO2R online tool (https://www.ncbi.nlm.nih.gov/geo/geo2r/) (accessed on 2 May 2025), which utilizes the GEOquery and limma packages within R to process microarray data [18]. DEGs were selected based on an adjusted *p*-value threshold of < 0.01 (Benjamini–Hochberg corrected) and an absolute log fold change (|logFC|) greater than 1.0. InteractiVenn (https://www.interactivenn.net/) (accessed on 2 May 2025) was then used to identify common DEGs across the datasets [19].

### 2.3. Hallmark Enrichment Analysis

To explore the functional relevance of overlapping differentially expressed genes (DEGs) from four GEO datasets (GSE32863, GSE27262, GSE40275, and GSE33356), we performed hallmark enrichment analysis using the Cancer Hallmarks Analytics Tool (https://cancerhallmarks.com) on 3 May 2025. This platform maps input genes to ten curated cancer hallmark categories using a hypergeometric test against a known reference set [20]. DEGs were uploaded using official gene symbols, and default parameters were applied. Statistically significant enrichment was determined based on adjusted *p*-values (FDR < 0.05). Results were visualized through a Hallmark Enrichment Plot comprising two layers: the outer ring displays hallmark categories significantly enriched in the DEG set, where the size and color intensity of each slice reflect the enrichment magnitude; only statistically significant hallmarks are highlighted. The inner ring compares gene distribution across hallmarks, where black dots represent hallmarks with gene counts exceeding expectation (positioned closer to the rim), while grey dots indicate no enrichment. This visualization provided a comprehensive overview of the biological processes significantly associated with LUAD progression in our dataset.

### 2.4. Gene Ontology and KEGG Pathway Analysis

For gene set enrichment analysis, ShinyGo v0.82 (http://bioinformatics.sdstate.edu/go/) (accessed on 23 April 2025) was used to perform both Gene Ontology (GO) and Kyoto Encyclopedia of Genes and Genomes (KEGG) analyses [21,22,23]. This allowed us to categorize DEGs based on cellular components, biological processes, and molecular functions and to identify associated signaling pathways. To define statistical significance, the following thresholds were applied consistently across all enrichment analyses: *p*-value < 0.01 and false discovery rate (FDR) < 0.05. These cut-offs ensured the robustness of enriched term identification. ShinyGO ranks terms based on FDR values, gene count, and fold enrichment score to elucidate the cellular functions and pathways engaged by the DEGs.

### 2.5. Protein–Protein Interaction Network

The STRING database (http://string-db.org/) (accessed on 2 May 2025) was used to construct a protein–protein interaction (PPI) network. Significant interactions were identified using a lower threshold of 0.4 for the combined score [24,25]. Visualization and further analysis of the PPI network were performed using Cytoscape v3.10.2. The CytoHubba plugin was employed to identify the top 10 hub genes based on their connectivity, and the MCODE plugin (version 1.5.1) (Degree Cut-off = 2, Node Score Cut-off = 0.2, K-Core = 2, and Max. Depth = 100) was used to detect significant clusters within these hub genes [26,27]. To prioritize key nodes within the network, we employed the CytoHubba plugin in Cytoscape, which provides a suite of topological analysis algorithms for hub gene identification [26,27]. To identify the key hub genes from the protein–protein interaction network, we used the cytoHubba plugin in the Cytoscape software, applying the Maximal Clique Centrality (MCC) algorithm, which has been shown to be one of the most effective methods for detecting essential nodes in biological networks [27]. The top 10 hub genes—namely, *AURKA*, *TOP2A*, *CCNB2*, *CENPF*, *MCM4*, *TPX2*, *KIF20A*, *ASPM*, *MELK*, and *NEK2*—were selected for further downstream analysis, including mRNA and protein expression level and prognostic evaluation.

### 2.6. Genomic Expression and Prognostic Value in LUAD

The Kaplan–Meier plotter (https://kmplot.com/analysis/) (accessed on 2 May 2025) was utilized to evaluate the prognostic significance of the expression levels of these genes in LUAD [28,29]. Additionally, UALCAN (https://ualcan.path.uab.edu/index.html) (accessed on 2 May 2025) and the Lung Cancer Explorer (https://lce.biohpc.swmed.edu/lungcancer/index.php#about) (accessed on 2 May 2025) were used to provide meta-analytical insights and comparative and correlation analysis, thereby enhancing the objectivity of our findings.

### 2.7. Immunohistochemistry and Transcriptomic Analysis

To evaluate the protein expression level of MCM4 in human tissues, immunohistochemical (IHC) staining data from normal and lung adenocarcinoma (LUAD) tissues were retrieved from the Human Protein Atlas (HPA; https://www.proteinatlas.org/) (accessed on 2 May 2025). The staining intensity and localization were visually compared to assess differences in MCM4 protein expression between normal lung epithelium and LUAD tissues. In addition, the TNMplot platform (https://tnmplot.com/analysis/) (accessed on 2 May 2025) was utilized to investigate the differential mRNA expression levels of *MCM4* across normal tissues, primary lung tumors, and metastatic lung tumors. To statistically compare *MCM4* expression among these three groups, the Kruskal–Wallis test, a non-parametric method for comparing median values across multiple independent groups, was applied. This test was chosen due to its robustness in analyzing non-normally distributed gene expression data.

### 2.8. Statistical Analysis

The two-tailed *t*-test was used to compare mRNA expression levels between LUAD and normal tissue samples from TCGA. To adjust for multiple hypothesis testing, we applied the Benjamini–Hochberg procedure. Survival analyses were conducted using Kaplan–Meier plots and log-rank tests, complemented by univariate Cox regression.

## 3. Results

### 3.1. Deciphering the Differential Gene Expression Landscape in Lung Cancer Through Public Data

We analyzed four microarray datasets from the GEO repository to identify DEGs in lung cancer: GSE32863, GSE27262, GSE40275, and GSE33356. These datasets were divided into normal and cancerous lung tissue samples. Volcano plots were employed to visualize the DEGs, with an adjusted *p*-value threshold set to <0.01 for statistical significance. The volcano plots were represented as the negative logarithm base 10 of the *p*-value (−log10(*p*-value)) and plotted against the log2 fold change to quantify changes in expression levels (Figure 2A–D). From the GSE32863 dataset, which includes 58 normal and 58 tumor specimens, we identified 514 genes with elevated expression and 756 genes with reduced expression in cancerous tissues. In the GSE27262 dataset, which includes 25 pairs of normal and cancerous tissues, we identified 909 genes with increased and 1364 genes with decreased expression. In the GSE40275 dataset, which includes 43 normal and 8 tumor samples, we identified 3010 upregulated and 2461 downregulated genes. The analysis of GSE33356, which includes 60 normal and 60 tumor tissues, revealed 455 upregulated and 949 downregulated genes. We then identified the commonly dysregulated DEGs among the four datasets. A Venn diagram was used to determine the overlapping DEGs, revealing 333 genes (Figure 2E–H). Among these, 68 were commonly upregulated (Figure 2F), and 255 were commonly downregulated (Figure 2G). Notably, the sum of commonly upregulated and downregulated genes (68 + 255 = 323) is fewer than the total 333 DEGs reported in Figure 2H. This discrepancy arises because Figure 2F,G only includes genes that are consistently upregulated or downregulated across all four datasets, while Figure 2H presents the total overlapping DEGs regardless of expression direction. Some genes may not meet the strict criteria for consistent up- or downregulation but are still significantly differentially expressed (FDR < 0.05, |log2FC| > 1) in at least one dataset. These genes were excluded from Figure 2F,G due to directionality inconsistency but are included in Figure 2H and subsequent analyses for comprehensive coverage. Identifying this intersection of DEGs is a critical step in pinpointing genes that could play a pivotal role in the pathogenesis of lung cancer, laying the groundwork for subsequent investigations into their functional roles and therapeutic potential.

### 3.2. Functional Enrichment Analysis Reveals Metastasis-Associated Hallmarks and Pathways in LUAD

To explore the biological significance of the intersectional differentially expressed genes (DEGs) identified from the four GEO datasets, we first conducted a hallmark enrichment analysis using the Cancer Hallmark Analysis Tool. The DEGs were significantly enriched in hallmark categories related to immune escape (adjusted *p* = 0.00115; OR = 2.11), angiogenesis (adjusted *p* = 0.00009; OR = 2.34), and invasion and metastasis (adjusted *p* = 0.00009; OR = 1.82) (Figure 3A,B). These findings indicate that the DEGs are highly involved in mechanisms underlying tumor progression, particularly metastatic dissemination in LUAD.

To further investigate the molecular functions of these genes, we performed Gene Ontology (GO) and Kyoto Encyclopedia of Genes and Genomes (KEGG) enrichment analyses. GO Biological Process (BP) analysis revealed that DEGs were significantly enriched in pathways associated with vascular development, including angiogenesis, vasculature morphogenesis, tube formation, and circulatory system development. Additional enrichment was observed in the regulation of cell proliferation, cell adhesion, organ morphogenesis, and tissue development (Figure 3C), which are all fundamental to tumor expansion and metastasis. In the Cellular Component (CC) category, DEGs were primarily associated with the extracellular matrix (ECM) and related structures, including collagen-containing ECM, cell–cell junctions, plasma membrane microdomains, secretory granules, extracellular vesicles, and exosomes (Figure 3D). These structures play pivotal roles in the remodeling of the tumor microenvironment and facilitate metastatic progression. Molecular Function (MF) analysis highlighted significant enrichment in receptor-mediated and structural binding activities, particularly TGF-β–activated receptor activity, integrin binding, cytokine binding, and ECM structural components (Figure 3E), supporting the hypothesis that these DEGs participate in pathways central to immune evasion and cell motility. KEGG pathway analysis further corroborated these results, revealing enrichment in several metastasis-related pathways, including ECM–receptor interaction, complement and coagulation cascades, leukocyte transendothelial migration, fluid shear stress and atherosclerosis, and the relaxin signaling pathway (Figure 3F). These signaling axes are known to regulate cancer cell invasion, immune modulation, and vascular dissemination. Collectively, the functional enrichment analyses demonstrated that the identified DEGs are not only key players in lung cancer pathogenesis but also strongly associated with hallmarks of metastasis. Their involvement in angiogenesis, ECM remodeling, cell adhesion, and TGF-β signaling highlights their potential as prognostic biomarkers or therapeutic targets in LUAD progression.

### 3.3. Identification of LUAD-Associated Hub Genes Through PPI Network Construction and Centrality Analysis

To elucidate the molecular interactions among the differentially expressed genes (DEGs) in LUAD, we constructed a protein–protein interaction (PPI) network using the STRING database, based on 333 DEGs filtered by a false discovery rate (FDR) < 0.05 and a minimum interaction confidence score > 0.01. The resulting network comprised 332 nodes and 1181 edges, with a highly significant PPI enrichment *p*-value < 1.0 × 10^−16^, indicating a non-random and biologically meaningful set of interactions. To identify genes that are most central to the structure and function of this network, we applied the cytoHubba plugin within Cytoscape, utilizing the Maximal Clique Centrality (MCC) algorithm—a method known for its high sensitivity in detecting essential nodes in biological networks. This analysis prioritized the top 10 hub genes, including *AURKA* (Aurora kinase A), *TOP2A* (DNA topoisomerase II α), *CCNB2* (Cyclin B2), *TPX2* (Targeting protein for Xklp2), *MCM4* (Minichromosome maintenance complex component 4), *KIF20A* (Kinesin family member 20A), *CENPF* (Centromere protein F), *ASPM* (Assembly factor for spindle microtubules), *NUSAP1* (Nucleolar and spindle-associated protein 1), *MELK* (Maternal embryonic leucine zipper kinase), and *NEK2* (NIMA-related kinase 2) (Figure 4A). To further validate the relevance of these hub genes in lung adenocarcinoma, we examined their expression profiles in TCGA LUAD datasets. All ten hub genes were significantly upregulated in LUAD tissues compared to normal controls (Figure 4B–K). Additionally, this expression trend was consistently observed across four independent GEO datasets, reinforcing the robustness of these findings. These results highlight a panel of functionally interconnected genes that are not only overexpressed in LUAD but are also structurally central within the LUAD-specific interaction network. Their prominent roles in mitotic regulation, DNA replication, and cell cycle progression suggest that they may serve as potential biomarkers or therapeutic targets for LUAD.

### 3.4. Prognostic Value of Hub Genes and MCM4 in Overall and Post-Progression Survival of LUAD Patients

To assess the prognostic significance of the identified hub genes in LUAD, we conducted survival analysis using the Kaplan–Meier plotter platform. The results demonstrated that high expression levels of all ten hub genes were significantly associated with shorter overall survival (OS) in LUAD patients, with hazard ratios (HRs) ranging from 1.54 to 2.35 (Figure 5). These findings suggest that the elevated expression of these genes may contribute to more aggressive disease progression and worse patient outcomes. Among the hub genes, *MCM4* showed particularly strong prognostic relevance. In the AJCC N2 subgroup—representing LUAD patients with advanced lymph node metastasis—high *MCM4* expression was associated with an even higher risk of death, with an HR of 2.53 (Figure 6). Furthermore, in the post-progression survival (PPS) analysis, which measures survival after disease progression, *MCM4* remained significantly associated with poor outcome (HR = 1.78, Figure 7). These findings indicate that, beyond its association with overall survival, *MCM4* may also predict disease aggressiveness and clinical deterioration following progression. Its elevated expression in both primary tumors and metastatic contexts underscores its potential as a prognostic biomarker and possibly a therapeutic target in advanced-stage LUAD.

### 3.5. MCM4 Is Highly Expressed in LUAD and Correlates with Tumor Progression, Metastasis, and Patient Survival

To further investigate the expression dynamics and clinical relevance of *MCM4* in non-small-cell lung cancer (NSCLC), we analyzed multiple datasets and platforms. Using the Lung Cancer Explorer (LCE) portal, *MCM4* was found to be significantly overexpressed in both LUAD (SMD = 1.79) and LUSC (SMD = 3.68), suggesting its relevance across NSCLC subtypes (Figure 8A). In our datasets GSE32863, GSE33356, and GSE27262—which contain paired tumor and adjacent normal tissues—paired *t*-test analysis showed that *MCM4* expression was consistently elevated in tumor tissues (Figure 8B–D). Similarly, TCGA LUAD data revealed that *MCM4* mRNA levels were significantly higher in tumor tissues, particularly in advanced-stage (Stage II–IV) compared to early-stage (Stage I) samples (Figure 8E,F). Protein-level analysis using TCGA proteomics and IHC staining from the Human Protein Atlas further confirmed that MCM4 protein was highly expressed in LUAD tissues (Figure 8G–I). In terms of clinical outcomes, First Progression Survival (FPS) analysis of AJCC N2 subtype demonstrated that high *MCM4* expression was significantly associated with shorter progression-free survival (HR = 2.09, log-rank *p* = 1.2 × 10^−13^), suggesting its potential role in promoting tumor growth and contributing to poor treatment response in early stages (Figure 8J). To determine whether *MCM4* expression is linked to metastasis, we analyzed its expression across AJCC N staging subgroups in TCGA. *MCM4* was significantly upregulated in LUAD tissues with lymph node metastasis, and in the N2 subgroup, patients with high *MCM4* expression had markedly worse FPS outcomes (HR = 4.33, log-rank *p* = 7.6 × 10^−6^) (Figure 9A,B). We further validated this trend using the TNMplot platform, which showed that *MCM4* expression increased progressively from normal lung tissue to primary LUAD and metastatic LUAD, as determined by the Kruskal–Wallis test. Notably, the fold change from normal to primary tumor was 3.25, indicating early involvement of *MCM4* in tumorigenesis, while the additional increase in metastasis (fold change = 1.22) also reached statistical significance, supporting *MCM4’s* role in metastatic progression (Figure 9C). To further investigate the potential role of MCM4 in LUAD invasiveness, we analyzed its expression correlation with matrix metalloproteinases (MMPs), which are known to facilitate extracellular matrix degradation and tumor metastasis [30,31]. Using the Lung Cancer Explorer platform, we observed that *MCM4* expression was positively correlated with *MMP9* and *MMP12* across both TCGA and GSE32863 datasets, supporting a potential role of *MCM4* in promoting ECM degradation and invasion (Figure 9D,E). Interestingly, the correlation between *MCM4*, *MMP1*, and *MMP13* was weaker and less consistent. This variability may be attributable to LUAD subtype heterogeneity or tumor-specific regulatory differences. These results highlight *MCM4’s* selective association with key MMPs and underscore its potential involvement in the metastatic progression of LUAD.

## 4. Discussion

LUAD is the most common subtype of NSCLC and a leading cause of cancer-related mortality worldwide [2]. Despite substantial progress in surgical techniques, chemotherapy, radiotherapy, targeted therapy, and immunotherapy, the five-year survival rate for LUAD remains disappointingly low [3,32,33,34,35,36]. A primary contributor to this poor prognosis is the high incidence of metastasis at the time of diagnosis. Tumor metastasis not only complicates clinical management but also limits the effectiveness of current therapeutic regimens. This underscores the critical need for identifying novel biomarkers and therapeutic targets that can inform early detection and more precise intervention strategies. In the present study, we leveraged integrated bioinformatics analyses to investigate DEGs associated with LUAD progression. To better elucidate the molecular interactions among differentially expressed genes (DEGs), we constructed a protein–protein interaction (PPI) network using the STRING database. In this analysis, both upregulated and downregulated DEGs were included to provide a comprehensive understanding of LUAD-specific molecular networks. This integrative approach ensures that functionally relevant gene modules and central regulators (hub genes) are identified, regardless of the direction of their expression change. Such inclusion is crucial, as both up- and downregulated genes may play cooperative or antagonistic roles within the same biological processes, particularly in metastasis-related pathways. Through comprehensive screening and prioritization using multiple GEO and TCGA datasets, we identified a panel of 10 hub genes exhibiting significantly elevated expression in LUAD tissues relative to normal tissues. Among these, *MCM4* emerged as a particularly promising candidate due to its strong correlation with clinical aggressiveness and poor prognosis.

Within this framework, *AURKA*, a serine/threonine kinase integral to centrosome stability and mitotic spindle assembly, was identified as a gene of interest [37,38]. The Aurora kinase family, which includes *AURKA*, *AURKB*, and *AURKC*, has been implicated in various cancers, with overexpression linked to chromosomal amplification of the *AURKA* gene [39,40,41,42,43]. Elevated levels of *AURKA* are associated with aggressive cancer behaviors, including invasion, metastasis, and resistance to conventional therapies [44,45,46,47,48,49,50]. Abnormal expression of *AURKA* has been observed in breast, colorectal, gastric, pancreatic, lung, and ovarian cancers [51,52,53,54]. Encouragingly, AURKA inhibitors have shown significant potential in enhancing the efficacy of various existing therapies, underscoring the value of *AURKA* as a therapeutic target in LUAD and highlighting the benefits of combining AURKA inhibitors with existing drugs [55,56,57,58,59]. Thus, elucidating the role of *AURKA* in lung cancer and examining the gene and cellular functions it regulates are crucial for assessing its therapeutic potential. Using Cytoscape, we found that *AURKA* was the top-ranked hub gene in LUAD. In this study, *AURKA* was significantly and highly expressed in LUAD and was associated with shorter OS, PPS, and OS in the AJCC N2 subtype. These analysis results are consistent with the current published literature and highlight the importance of *AURKA*.

Similar to *AURKA*, significantly higher expression of *TOP2A* in LUAD was associated with shorter overall survival. *TOP2A* is a crucial enzyme that maintains DNA topology during transcription, replication, and chromosome segregation by creating transient double-stranded breaks in the DNA molecule [60]. Overexpression of *TOP2A* is associated not only with tumorigenesis and cancer progression in various types of cancer, such as breast, hepatocellular, colorectal, lung, and ovarian, but also with increased proliferation, angiogenesis, metastasis, and drug resistance [61,62,63]. However, when analyzing the OS of the AJCC N2 patient subgroup expressing *TOP2A*, there was no significant correlation between high expression of *TOP2A* and patient survival (log rank *p* = 0.087). The role of *TOP2A* in the metastatic progression of LUAD needs to be further clarified.

Our results showed that high expression of *CCNB2* was significantly correlated with OS, PPS, and OS in the AJCC N2 subtype in LUAD patients. CCNB2 is a member of the B-type cyclins and essential for cell cycle regulation, particularly in the G2/M transition, where it activates cyclin-dependent kinase 1. This regulation ensures proper cell division and genomic stability [64]. In cancer, *CCNB2* is often overexpressed, contributing to uncontrolled cell proliferation and tumorigenesis. Abnormal expression of *CCNB2* has been observed in various cancers, including breast, nasopharyngeal, clear cell renal cell carcinoma, and low-grade glioma [64,65,66,67]. Overexpression of *CCNB2* is associated with increased proliferation, angiogenesis, metastasis, and drug resistance [64,65,66]. Although *CCNB2* ranked third among the hub genes in our study, its significant PPS and OS in the AJCC N2 subtype in LUAD patients were consistent with the findings in other published literature and may have the potential to serve as a metastasis target.

*CENPF* is a crucial component of the kinetochore–centromere complex, ensuring proper chromosome alignment and segregation during mitosis [68,69]. In breast, adrenocortical, cervical, ovarian, and non-small-cell lung cancers, aberrant expression of *CENPF* has been observed [68,69,70,71,72]. *CENPF* overexpression is associated with increased proliferation, angiogenesis, metastasis, and drug resistance [68,69,70,71]. Although *CENPF* ranked fourth among hub genes, its high expression was only correlated with OS and OS in the AJCC N2 subtype, indicating limited potential as a metastatic driver gene.

*MCM4* is an essential component of the replicative helicase complex and plays a vital role in DNA replication and cell cycle progression [12]. In cancer, *MCM4* is frequently overexpressed, contributing to genomic instability and tumor progression. Abnormal expression of *MCM4* has been observed in various cancers, including glioma, hepatocellular carcinoma, colorectal cancer, and uterine corpus endometrial carcinoma [8,12,73]. Moreover, the overexpression of *MCM4* is associated with increased proliferation, angiogenesis, metastasis, invasion, and drug resistance [8,12,73]. Our findings indicate that *MCM4* is consistently overexpressed in LUAD at both the transcript and protein levels, as evidenced by data from GEO datasets, TCGA, and the Human Protein Atlas. Notably, *MCM4* expression was significantly higher in tumors with advanced stage and lymph node metastasis, and it increased progressively from normal lung tissue to primary tumors and further to metastatic LUAD. Survival analyses further revealed that high *MCM4* expression is associated with reduced overall survival and poor post-progression survival, particularly in patients with advanced nodal involvement (AJCC N2 subgroup). These observations strongly suggest that *MCM4* plays a central role in promoting LUAD progression and could serve as a predictive biomarker for disease aggressiveness and early relapse. Moreover, its consistent dysregulation across multiple datasets highlights *MCM4’s* potential as a robust and clinically relevant target. Taken together, this study provides compelling evidence that *MCM4* contributes to the metastatic phenotype of LUAD and holds promise as a diagnostic, prognostic, and therapeutic biomarker. Notably, we further observed that *MCM4* expression was positively correlated with the expression levels of several matrix metalloproteinases (*MMP1*, *MMP9*, *MMP12*, and *MMP13*) in both TCGA-LUAD and GSE31210 datasets. Given the established roles of MMPs in extracellular matrix degradation and cancer cell invasion, these correlations reinforce the notion that *MCM4* may promote LUAD progression through enhancing metastatic and invasive potential [30,31]. Targeting *MCM4* or its downstream pathways may represent a novel therapeutic approach, especially for patients with advanced-stage disease who currently lack effective treatment options. Further mechanistic studies and clinical validation are warranted to elucidate *MCM4’s* role in LUAD pathobiology and translate these findings into clinical practice.

*TPX2* is a critical regulator of spindle assembly and microtubule nucleation during mitosis, functioning closely with *AURKA*. Aberrant expression of *TPX2* has been reported in various human cancers and is often associated with aggressive tumor behavior [74,75]. In our study, *TPX2* was significantly upregulated in LUAD samples and strongly associated with poor OS and PPS, highlighting its potential role in LUAD progression. Previous studies have demonstrated that TPX2 promotes AURKA activation and downstream PI3K/AKT signaling, thereby facilitating angiogenesis in bladder cancer cells [76]. Furthermore, in breast and colorectal cancers, *TPX2* has been identified as a *MYC*-cooperating oncogene that drives tumorigenesis [41]. Notably, *TPX2* and *AURKA* often exhibit co-overexpression and functional interdependence in several malignancies. TPX2 has been shown to stabilize AURKA and enhance its kinase activity, reinforcing their oncogenic synergy [75]. This is consistent with our observation that both *TPX2* and *AURKA* are highly expressed in LUAD and associated with adverse prognosis. Taken together, these results support the notion that TPX2, potentially in coordination with AURKA, plays a key role in driving LUAD malignancy and may serve as a prognostic biomarker or therapeutic target.

Similarly, overexpression of *KIF20A* is associated with increased proliferation, angiogenesis, metastasis, and drug resistance and further contributes to tumor progression and poor prognosis [77,78]. *KIF20A* is a kinesin family member, playing a critical role in mitosis by transporting chromosomes and ensuring proper spindle assembly and cytokinesis [77]. Abnormal expression of *KIF20A* has been observed in various cancers, including breast, pancreatic, glioma, hepatocellular carcinoma, esophageal squamous cell carcinoma, bladder, and prostate cancers [77,78]. Although *KIF20A* did not have the highest HR and the most significant association with short survival in multiple KM plot analyses, the relationship between *KIF20A* and malignant progression and metastasis in LAUD patients still deserves further exploration in future studies.

A centrosomal protein crucial for regulating the mitotic spindle, *ASPM* ensures proper chromosome alignment and segregation during cell division [79]. Initially identified for its role in neurogenesis and brain size regulation, *ASPM* is frequently overexpressed in various cancers, including pancreatic ductal adenocarcinoma, gastric cancer, small cell lung cancer, kidney renal clear cell carcinoma, liver hepatocellular carcinoma, and gliomas [79,80,81], and is also associated with increased proliferation, angiogenesis, metastasis, and drug resistance [80,81,82]. For example, high levels of *ASPM* correlate with poor survival and aggressive tumor behavior in pancreatic ductal adenocarcinoma as well as with higher tumor grades and recurrence in gliomas [79,81]. *ASPM* ranked eighth among hub genes. Although *ASPM* did not have the highest HR and the most significant association with short survival in multiple KM plot analyses, the relationship between *ASPM* and malignant progression and metastasis in LAUD patients still deserves further exploration in future studies.

*MELK* is a serine/threonine kinase involved in multiple cancer-related processes, including cell cycle regulation, apoptosis suppression, and stem cell maintenance [83,84]. Overexpression of *MELK* has been documented in several malignancies and is often associated with tumor proliferation, invasion, and treatment resistance. In hepatocellular carcinoma (HCC), recent studies have highlighted *MELK’s* potential oncogenic role. For instance, *MELK* has been shown to promote HCC cell migration and invasion via activation of the FOXM1 pathway, suggesting its involvement in metastasis [85]. In our current study, *MELK* was identified among the top 10 hub genes through protein–protein interaction network analysis. Its mRNA expression was significantly elevated in LUAD tissues compared to adjacent normal lung tissues, a finding consistently validated across TCGA and multiple GEO datasets. However, despite its overexpression, *MELK* did not show a statistically significant correlation with post-progression survival (PPS) in our cohort (log-rank *p* = 0.085). This contrasts with genes such as *MCM4*, whose elevated expression was strongly associated with poor prognosis in both overall survival and PPS analyses. Therefore, although *MELK* may participate in LUAD tumorigenesis and has been highlighted in prior studies, our results do not support its role as a primary prognostic indicator of disease aggressiveness or late-stage deterioration. Consequently, we did not pursue further mechanistic or functional analyses of *MELK* in this study. These findings underscore the importance of validating putative biomarkers across different clinical endpoints. *MELK* may still represent a context-dependent oncogenic factor in LUAD, particularly in subtypes or treatment settings not captured in our current analysis. Further research is needed to clarify its role and therapeutic relevance in specific LUAD subgroups.

Finally, abnormal expression of *NEK2* has been observed in various cancers, including lung, pancreatic, prostate, and breast, as well as hepatocellular carcinoma and multiple myeloma [86,87,88]. NEK2 is a serine/threonine kinase crucial for cell cycle regulation, especially in centrosome duplication and spindle assembly during mitosis; these processes ensure proper chromosome segregation and genomic stability. *NEK2* ranked tenth among hub genes. The overexpression of *NEK2* in cancer leads to chromosomal instability and tumor progression and is associated with increased proliferation, angiogenesis, metastasis, and drug resistance [86,87,88]. In lung cancer, high levels of *NEK2* correlate with poor prognosis and aggressive disease [87,88]. Surprisingly, in multiple KM plot analyses, *NEK2* did not have the highest HR and the most significant correlation with short-term survival. The relationship between the *NEK2*-promoting metastasis mechanism and the genes involved needs further exploration.

Given its consistent upregulation in tumor tissues, significant correlation with advanced disease features, and strong association with poor survival outcomes, *MCM4* emerges as a promising candidate for further functional investigation. Future studies should focus on elucidating the molecular mechanisms by which *MCM4* contributes to LUAD metastasis, potentially involving regulation of extracellular matrix remodeling, cell cycle progression, or immune evasion. Experimental strategies such as CRISPR-Cas9-mediated gene editing, pathway inhibition assays, and in vivo metastasis models could help clarify its pathogenic role. Moreover, considering its strong prognostic value, *MCM4* may serve as a potential therapeutic target. Small molecule inhibitors or antibody-based approaches targeting *MCM4* could be explored to assess therapeutic efficacy, especially in patients with metastatic LUAD.

## 5. Conclusions

In this study, we integrated multiple transcriptomic datasets and conducted comprehensive bioinformatics analyses to uncover genes associated with lung adenocarcinoma (LUAD) metastasis. Our findings highlight *MCM4* as a potential metastatic biomarker and prognostic indicator for LUAD, warranting further mechanistic and therapeutic investigation.

## Figures and Tables

**Figure 1 diagnostics-15-01555-f001:**
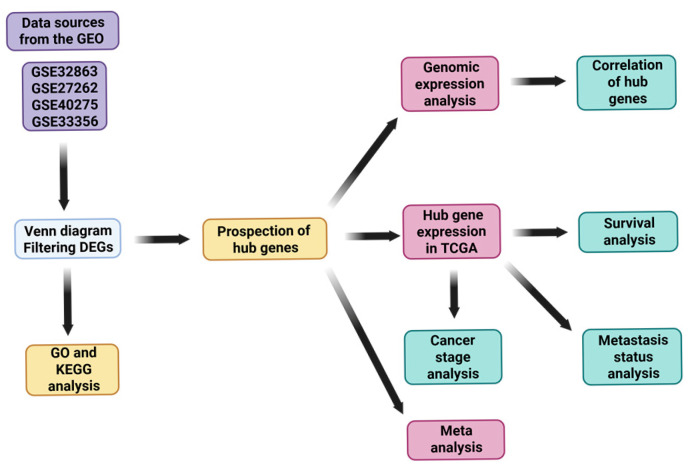
Flowchart of this study.

**Figure 2 diagnostics-15-01555-f002:**
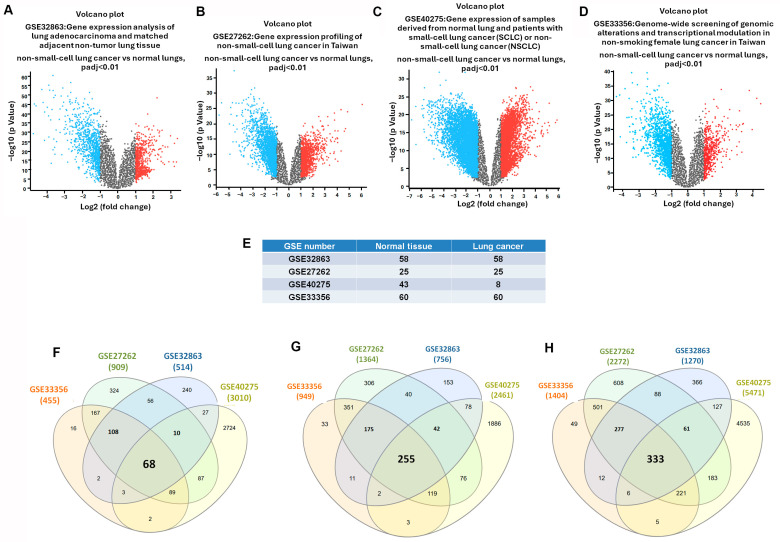
Intersecting DEGs in GEO datasets of lung cancer: (**A**–**D**) volcano plots of genes significantly upregulated (red) and downregulated (blue) in lung tumors compared to normal tissue; (**E**) sample sizes: GSE32863 (58 normal and 58 lung cancer tissues), GSE27262 (25 normal and 25 lung cancer tissues), GSE40275 (43 normal and 8 lung cancer tissues), and GSE33356 (60 normal and 60 lung cancer tissues); (**F**–**H**) Venn diagrams of overlapping upregulated, downregulated, and all DEGs.

**Figure 3 diagnostics-15-01555-f003:**
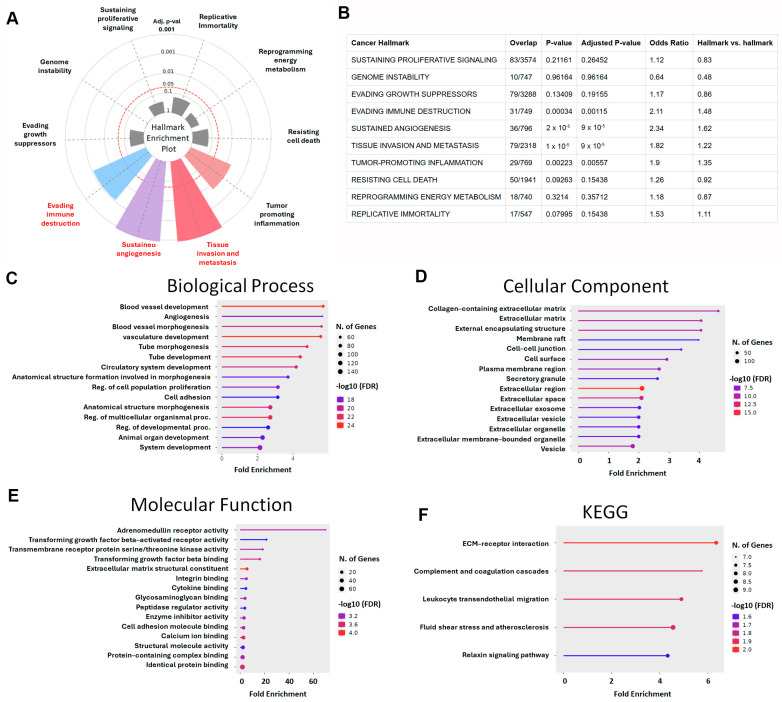
Functional enrichment analysis of LUAD-associated DEGs: (**A**) Hallmark Enrichment Plot displaying the distribution of DEGs across ten canonical cancer hallmarks. Colored slices represent significantly enriched hallmarks (adjusted *p* < 0.05), and slice size indicates enrichment strength. (**B**) This table summarizes the enrichment of LUAD-associated differentially expressed genes (DEGs) across ten classical cancer hallmarks using the Cancer Hallmarks Analytics Tool. Columns include cancer hallmark, overlap (DEGs/reference), *p*-value, adjusted *p*-value, odds ratio, and hallmark vs. hallmark ratio. (**C**) Gene Ontology (GO) biological process enrichment analysis of DEGs. (**D**) GO cellular component enrichment analysis. (**E**) GO molecular function enrichment analysis. (**F**) KEGG pathway enrichment analysis of DEGs (color marks false discovery rate).

**Figure 4 diagnostics-15-01555-f004:**
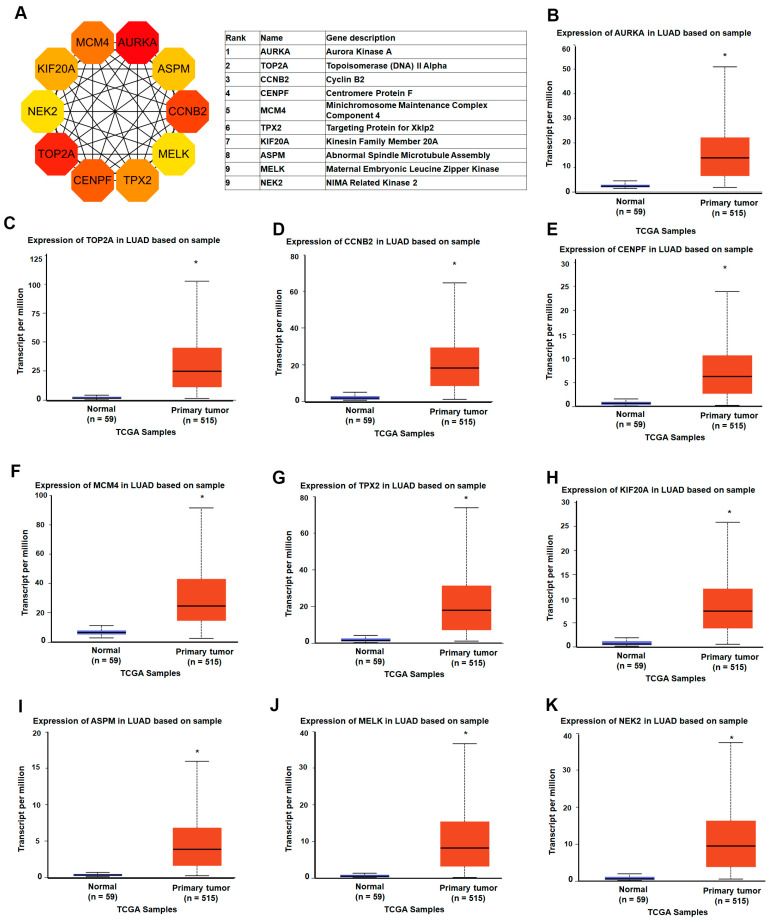
Identification and expression analysis of LUAD-associated hub genes: (**A**) the top 10 hub genes were identified using the Maximal Clique Centrality (MCC) algorithm in the cytoHubba plugin within Cytoscape; (**B**–**K**) gene expression analysis of the top 10 hub genes (*AURKA*, *TOP2A*, *CCNB2*, *TPX2*, *MCM4*, *KIF20A*, *CENPF*, *ASPM*, *NUSAP1*, and *NEK2*) in lung adenocarcinoma (LUAD) and normal lung tissues based on TCGA RNA-seq data. * *p*-value < 0.05.

**Figure 5 diagnostics-15-01555-f005:**
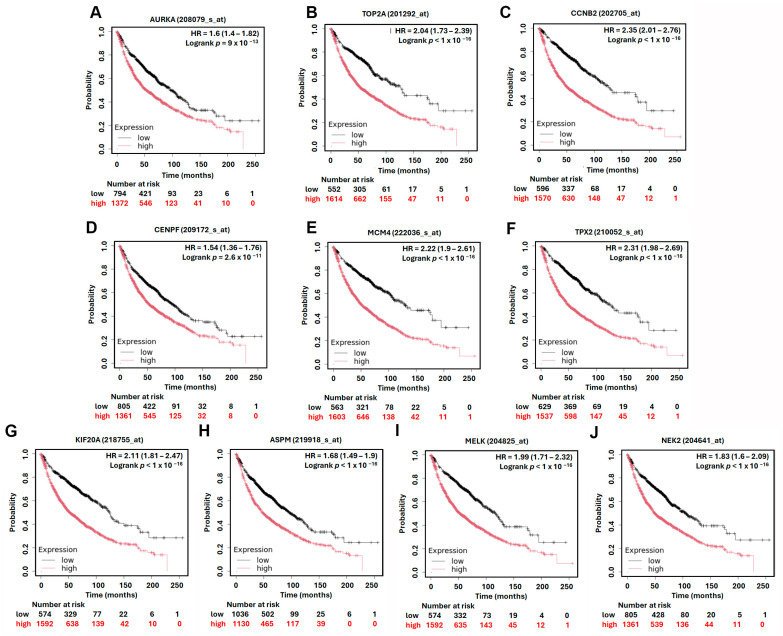
Kaplan–Meier overall survival analysis of LUAD patients based on the expression of top 10 hub genes. High and low expression groups are represented by red and black lines, respectively. Hazard ratios (HRs) and log-rank *p*-values are indicated in each plot. (**A**) *AURKA*, (**B**) *TOP2A*, (**C**) *CCNB2*, (**D**) *CENPF*, (**E**) *MCM4*, (**F**) *TPX2*, (**G**) *KIF20A*, (**H**) *ASPM*, (**I**) *MELK*, (**J**) *NEK2*.

**Figure 6 diagnostics-15-01555-f006:**
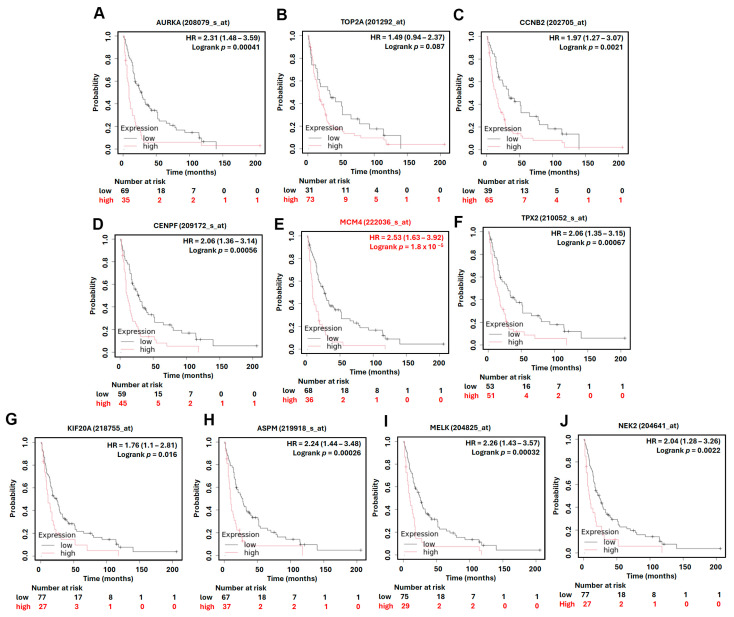
Kaplan–Meier overall survival in AJCC N2 subtype analysis of LUAD patients based on the expression of top 10 hub genes. High and low expression groups are represented by red and black lines, respectively. Hazard ratios (HRs) and log-rank *p*-values are indicated in each plot. (**A**) *AURKA*, (**B**) *TOP2A*, (**C**) *CCNB2*, (**D**) *CENPF*, (**E**) *MCM4*, (**F**) *TPX2*, (**G**) *KIF20A*, (**H**) *ASPM*, (**I**) *MELK*, (**J**) *NEK2*.

**Figure 7 diagnostics-15-01555-f007:**
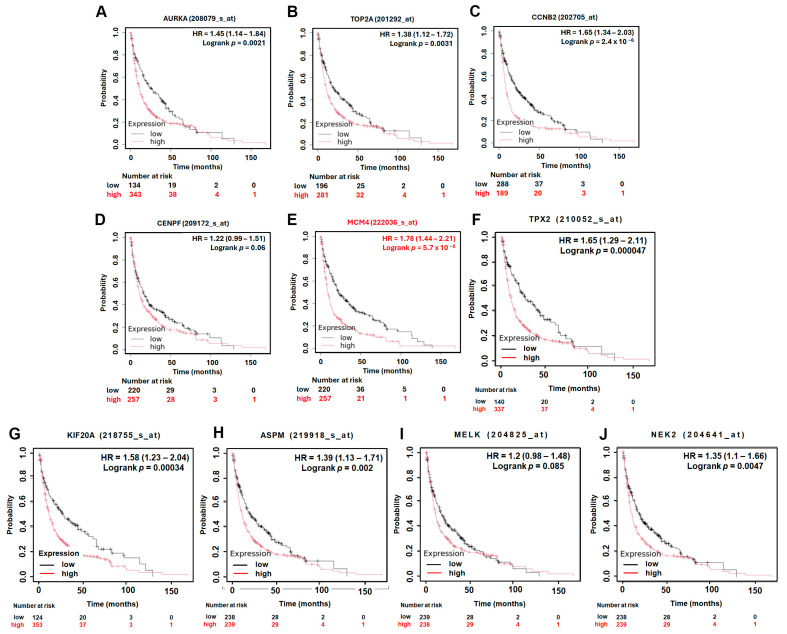
Kaplan–Meier post-progression survival (PPS) analysis of LUAD patients based on the expression of top 10 hub genes. High and low expression groups are represented by red and black lines, respectively. Hazard ratios (HRs) and log-rank p-values are indicated in each plot. (**A**) *AURKA*, (**B**) *TOP2A*, (**C**) *CCNB2*, (**D**) *CENPF*, (**E**) *MCM4*, (**F**) *TPX2*, (**G**) *KIF20A*, (**H**) *ASPM*, (**I**) *MELK*, (**J**) *NEK2*.

**Figure 8 diagnostics-15-01555-f008:**
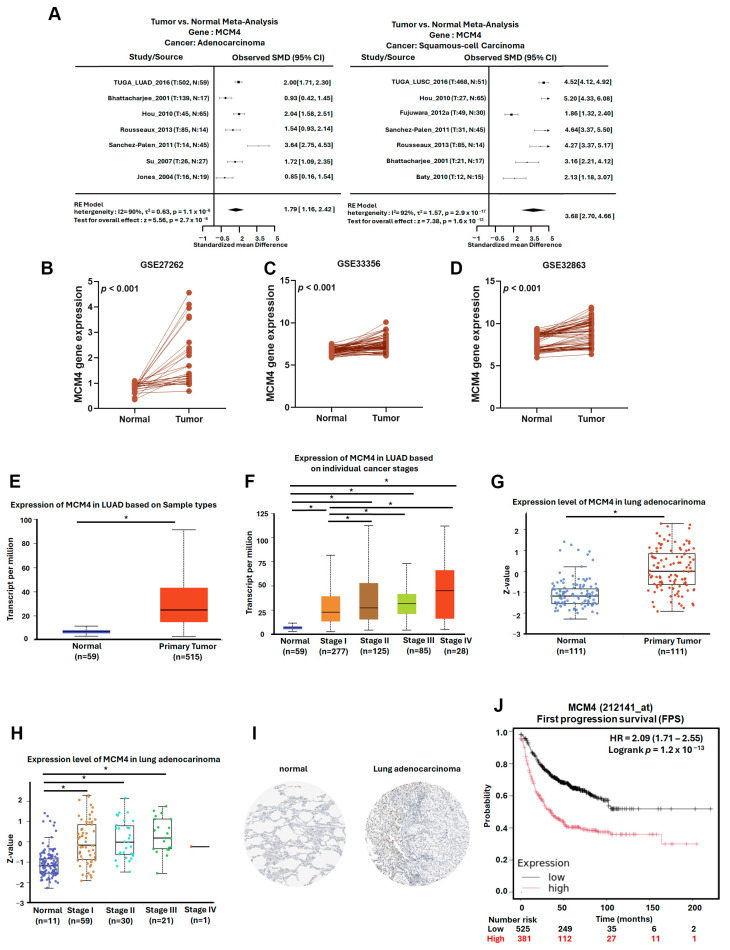
*MCM4* is highly expressed in LUAD and correlates with tumor progression and early poor prognosis: (**A**) meta-analysis of *MCM4* expression in LUAD and LUSC compared to normal lung tissues, obtained from the Lung Cancer Explorer (LCE); (**B**–**D**) paired t-test analysis of *MCM4* expression in tumor and adjacent normal tissues from GEO datasets GSE32863, GSE33356, and GSE27262; (**E**,**F**) *MCM4* expression in LUAD across different clinical stages (stage I–IV) using TCGA data; (**G**,**H**) MCM4 protein expression in LUAD and normal tissues based on TCGA proteomic analysis; (**I**) immunohistochemistry (IHC) staining results from the Human Protein Atlas demonstrating MCM4 protein expression in LUAD and normal lung tissues; (**J**) KM plot showing First Progression Survival (FPS) in LUAD patients stratified by *MCM4* expression level. * *p*-value < 0.05.

**Figure 9 diagnostics-15-01555-f009:**
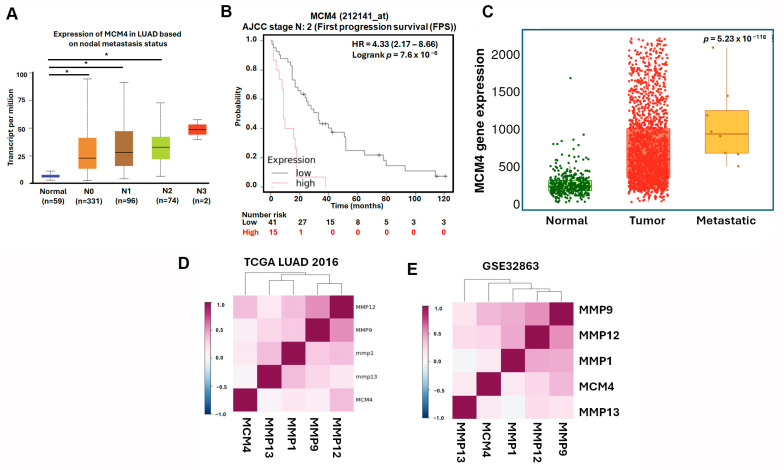
MCM4 expression is associated with lymph node metastasis and adverse survival outcomes in advanced LUAD: (**A**) *MCM4* mRNA expression levels in LUAD samples stratified by lymph node metastasis status; (**B**) KM plot of (First Progression Survival (FPS) in LUAD patients with AJCC N2 stage; and (**C**) *MCM4* expression analysis from the TNMplot platform across normal lung, primary LUAD, and metastatic LUAD tissues. Statistical significance was assessed using the Kruskal–Wallis test. (**D**,**E**) Correlation analysis of *MCM4* and matrix metalloproteinases (*MMP1*, *MMP9*, *MMP12*, and *MMP13*) expression levels in LUAD using Lung Cancer Explorer. * *p*-value < 0.05.

## Data Availability

The data presented in this study are available on request from the corresponding author.

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
