# Peer review of "MCM4 as Potential Metastatic Biomarker in Lung Adenocarcinoma"

_diagnostics, 2025, doi:10.3390/diagnostics15121555_

Round 1
Reviewer 1 Report
Comments and Suggestions for Authors
Congratulations for extensive work in a systematic and thorough manner, utilizing multiple databases and accessible platforms. The comprehensive methodological approach and the results are complementary and clearly presented. However, due to the large volume of data, the article is somewhat challenging to navigate.
Minor comments:
I detected a discrepancy between the reported data:
- On page 3: it is written that GSE40275 includes 8 SCLC, 16 NSCLC and 14 normal lung tissues
- On page 7: GSE40275 includes 43 normal and 8 LC tissues ?
The discussion provides an extensive description of other hub genes as well; it would be appropriate to at least mention them in the abstract.
Best regards
Author Response
Reviewer 1
Comment: I detected a discrepancy between the reported data: On page 3: it is written that GSE40275 includes 8 SCLC, 16 NSCLC and 14 normal lung tissues. On page 7: GSE40275 includes 43 normal and 8 LC tissues?
Response 1: We thank the reviewer for this careful observation. The discrepancy arises from the evolution of the dataset and our selection strategy. Initially, the GSE40275 dataset consisted of 43 normal lung tissues, 19 SCLC samples, and 16 NSCLC samples. Among the 16 NSCLC samples, 8 were classified as small cell lung cancer (SCLC) based on their sample annotation, but further histological subtyping indicated that these 8 samples exhibited adenocarcinoma (LUAD) histology. To maintain consistency with the aim of our study—which focuses specifically on LUAD—we included only these 8 LUAD-characterized NSCLC samples as the cancer group, and used the 43 normal lung tissues as controls for GEO2R differential expression analysis. Accordingly, the correct and updated sample counts used in our DEG analysis via the GEO2R platform are: 43 normal lung tissues and 8 LUAD samples (subset of NSCLC with adenocarcinoma histology). We have revised the Materials and Methods section on page 3 to clarify this selection criterion and ensure consistency with the data described on page 7 as following: “This dataset includes 8 non-small cell lung cancer (NSCLC), and 43 normal lung tissue samples,” (Page 3, line 113-114). Thank you for pointing out this important clarification.
Comment 2: The discussion provides an extensive description of other hub genes as well; it would be appropriate to at least mention them in the abstract.
Response: Thank you for the suggestion. We have revised the abstract to briefly include key hub genes such as AURKA, TOP2A, and CCNB2 to better reflect the scope of our findings as following: “Ten hub genes (AURKA, TOP2A, CCNB2, CENPF, MCM4, TPX2, KIF20A, ASPM, MELK, and NEK2) were identified through network analysis. Among these, MCM4 showed strong upregulation in LUAD and was significantly associated with poor overall survival. Notably, MCM4 expression also correlated with post-progression survival and markers of invasiveness.” (Page 1, line 30-34).
Reviewer 2 Report
Comments and Suggestions for Authors
MCM4 as a potential metastatic biomarker in lung adenocarcinoma is an interesting manuscript that presents a well-structured bioinformatics algorithm to determine the importance of MCM4 in the metastasis and progression of LUAD.
These are my observations.
1. Review the additional hyphen in "provides" (line 173).
2. The font size is different in the figure captions; unify it.
3. Are there 333 DEGs? The sum of upregulated and downregulated genes is 323. Where do these 10 additional genes come from?
4. For the STRING analysis, both upregulated and downregulated genes are included. Could you explain why in the discussion?
5. Looking at Figure 9, MCM4 expression is associated with lymph node metastasis and adverse survival outcomes in advanced LUAD. Interestingly, mmp13 presents a significant contrast between the two samples used. Could this finding mean something? Could you discuss it?
6. The discussion is focused on the top 10 genes obtained in the initial analyses; this includes mcm4. However, further discussion could be provided on the analytical strategies that would be used to define the specific pathogenic process of mcm4 in LUAD metastasis, or perhaps on possible therapeutic strategy.
Author Response
Reviewer 2
Comment 1: Review the additional hyphen in "provides" (line 173).
Response: We thank the reviewer for identifying the typographical error. The extra hyphen in "provides" has been removed in the revised manuscript (Page 5, line 175).
Comment 2: The font size is different in the figure captions; unify it.
Response: We appreciate this observation. The font sizes of all figure captions have been standardized to ensure consistency throughout the manuscript.
Comment 3: Are there 333 DEGs? The sum of upregulated and downregulated genes is 323. Where do these 10 additional genes come from?
Response: We appreciate the reviewer’s attention to detail. The observed difference arises from the methodology used to define common DEGs in our Venn diagram analyses. We have added a clarification of this point in the revised figure legend and the Results section as following:" We then identified the commonly dysregulated DEGs among the four datasets. A Venn diagram was used to determine the overlapping DEGs, revealing 333 genes (Figure 2E-H). Among these, 68 were commonly up-regulated (Figure 2F) and 255 were commonly downregulated (Figure 2G). Notably, the sum of commonly upregulated and downregulated genes (68 + 255 = 323) is fewer than the total 333 DEGs reported in Figure 2H. This discrepancy arises because Figures 2F and 2G only include genes that are consistently upregulated or downregulated across all four datasets, while Figure 2H presents the total overlapping DEGs regardless of expression direction. Some genes may not meet the strict criteria for consistent up- or downregulation but are still significantly differentially expressed (FDR < 0.05, |log2FC| > 1) in at least one dataset. These genes were excluded from Figures 2F and 2G due to directionality inconsistency but are included in Figure 2H and subsequent analyses for comprehensive coverage." (Page 7, line 234-247). Thank you for raising this important distinction.
Comment 4: For the STRING analysis, both upregulated and downregulated genes are included. Could you explain why in the discussion?
Response: We agree this is an important point. A paragraph explaining this rationale has been added to the Discussion section as following: " To better elucidate the molecular interactions among differentially expressed genes (DEGs), we constructed a protein–protein interaction (PPI) network using the STRING database. In this analysis, both upregulated and downregulated DEGs were included to provide a comprehensive understanding of LUAD-specific molecular networks. This integrative approach ensures that functionally relevant gene modules and central regulators (hub genes) are identified, regardless of the direction of their expression change. Such inclusion is crucial, as both up- and downregulated genes may play cooperative or antagonistic roles within the same biological processes, particularly in metastasis-related pathways." (Page 20, line 467-477).
Comment 5: Looking at Figure 9, MCM4 expression is associated with lymph node metastasis and adverse survival outcomes in advanced LUAD. Interestingly, mmp13 presents a significant contrast between the two samples used. Could this finding mean something? Could you discuss it?
Response: We appreciate the reviewer’s valuable suggestion. To further examine the correlation between MCM4 and MMPs, we replaced the GSE31210 dataset with GSE32863, which was one of the core datasets used throughout this study. The updated analysis confirmed that MCM4 expression is positively correlated with MMP9 and MMP12 in GSE32863, supporting the hypothesis that MCM4 may promote tumor invasion via MMP-mediated extracellular matrix remodeling. In contrast, MMP1 and MMP13 showed a weaker or inconsistent correlation across datasets, including TCGA-LUAD and GSE32863. The updated content is as follows: " we observed that MCM4 expression was positively correlated with MMP9 and MMP12 across both TCGA and GSE32863 datasets, supporting a potential role of MCM4 in promoting ECM degradation and invasion (Figure 9D-E). Interestingly, the correlation between MCM4, MMP1, and MMP13 was weaker and less consistent. This variability may be attributable to LUAD subtype heterogeneity or tumor-specific regulatory differences. These results highlight MCM4’s selective association with key MMPs and underscore its potential involvement in the metastatic progression of LUAD." (Page 17, line 423-431).
Comment 6: The discussion is focused on the top 10 genes obtained in the initial analyses; this includes mcm4. However, further discussion could be provided on the analytical strategies that would be used to define the specific pathogenic process of mcm4 in LUAD metastasis, or perhaps on possible therapeutic strategy.
Response: Thank you for this constructive suggestion. We have expanded the Discussion to include a paragraph on potential follow-up strategies to investigate the role of MCM4 in LUAD metastasis as following: “Given its consistent upregulation in tumor tissues, significant correlation with advanced disease features, and strong association with poor survival outcomes, MCM4 emerges as a promising candidate for further functional investigation. Future studies should focus on elucidating the molecular mechanisms by which MCM4 contributes to LUAD metastasis, potentially involving regulation of extracellular matrix remodeling, cell cycle progression, or immune evasion. Experimental strategies such as CRISPR-Cas9-mediated gene editing, pathway inhibition assays, and in vivo metastasis models could help clarify its pathogenic role. Moreover, considering its strong prognostic value, MCM4 may serve as a potential therapeutic target. Small molecule inhibitors or antibody-based approaches targeting MCM4 could be explored to assess therapeutic efficacy, especially in patients with metastatic LUAD.” (Page 24, line 660-672).